# Determining Gains Acquired from Word Embedding Quantitatively using Discrete Distribution Clustering

## Abstract

Word embeddings have become widely-used in document analysis. A large number of models have been proposed, but the net gain these models can achieve expectably beyond the traditional bag-of-words based approaches remains undetermined. Our empirical studies, conducted from a nonparametric unsupervised perspective, reveal where and how word embeddings can contribute to document analysis. Our approach is based on a recent algorithmic advance in nonparametric clustering for empirical measures, which neither invents nor relies on any document vector representations. The new document clustering approach proposed in this work is easy to use and stably outperforms other existing methodologies on a variety of document-clustering tasks.

## 1 Introduction

Word embeddings, or word vectors, have been broadly adopted for document analysis (Mikolov et al., 2013b,a). A key appeal of word embedding methods is that they can be obtained from external large-scale corpus and then be easily utilized for different data. Before choosing word embeddings for data analysis, researchers must first consider how much extra gain can be brought from the "embedded" knowledge of words in comparison with that achieved by existing bag-of-words based approaches. Moreover, they must also consider how to quantify that gain. Such a preliminary evaluation is often necessary before any further decisions can be made about the data.

Answering such questions is important: almost every model used in practice exploits some basic representations — bag-of-words and word embeddings — for the sake of its computational tractability. Based on word embeddings, high-level models are designed for various tasks. Examples include entity representations, similarity measures, data manifolds, hierarchical structures, language models, and neural architectures. Therefore, it is important to investigate whether the gain or loss found in practice should be credited to the extra assumptions associated with those high-level models or to the use of basic word embeddings. As our experiments demonstrate, introducing these extra assumptions will make individual methods effective only if certain constraints are met. We will address this issue from an unsupervised perspective.

Our proposed clustering framework has several advantages. Instead of suppressing a document into a fixed-length vector feeding post-analysis, our framework uses the Wasserstein distance (or the Earth Mover's Distance, EMD) as a meta-distance to quantify the dissimilarity between two empirical nonparametric measures (or discrete distributions) over word embedding space (Wan, 2007; Kusner et al., 2015). Hence, it excludes any vector representation of the documents and sidesteps extra high-level assumptions, which is crucial and beneficial to evaluating the gain from basic word embeddings.

Our approach is intuitive and robust. the Wasserstein distance considers the cross-term relationship between different words in a *principled* fashion. As defined, the distance between two documents, say A and B, are the minimum cumulative cost that words from document A need to "travel" to match exactly the point cloud of document B. Here, the travel cost of a path between two words is their (squared) Euclidean distance in the word embedding space. Therefore, how much

benefit Wasserstein distance brings also depends on how the utilized word embedding space flattens document data as distributions, disentangling independent factors into different regions, subject to the interest of task.

While Wasserstein distance is well suited for document analysis, a major obstacle of EMD-based approaches is the high-magnitude computation involved, especially for the original D2-clustering method (Li and Wang, 2008) — an EMD-based clustering framework. The main technical hurdle is to compute the Wasserstein barycenter, which is a discrete distribution, efficiently for a given set of discrete distributions. Thanks to the recent advance of algorithms for efficient solving of Wasserstein barycenters, one can now perform document clustering in a nonparametric way by directly treating them as empirical measures over word embedding space (Cuturi and Doucet, 2014; Ye and Li, 2014; Benamou et al., 2015; Ye et al., 2017). In short, the intuition behind such a nonparametric framework is: By keeping D2-clustering to a minimal assumption set, we can achieve results of the highest possible clustering quality regardless the size and patterns of data. Obtaining highest quality clustering of unstructured text data merits consuming extra computational resources. Because it is a crucial step in many techniques and applications of natural language processing, such as cross-document co-reference resolution (Singh et al., 2011), document summarization (Radev et al., 2004; Wang and Li, 2010), retrospective events detection (Yang et al., 1998), and opinion mining (Zhai et al., 2011).

**Our contributions**. Our work has two main contributions. First, we create a basic tool of document clustering with mere hyper-parameters at scale. Our tool leverages the state-of-the-art numerical toolbox developed for optimal transport to achieve computational feasibility. Meanwhile, it gives state-of-the-art clustering performances across heterogeneous text data — an advantage over other methods in the literature. Second, with our tool, one can quantitatively inspect how well a word-embedding model can fit the data and how much gain or loss will be obtained compared to traditional bag-of-words models. Acquiring insights on these questions is valuable for document analysis beyond clustering. The exploration of D2-clustering for documents will provide a window for investigating these questions.

## 2 Related Work

In the original D2-clustering framework (Li and Wang, 2008), calculating Wasserstein barycenter involves solving a large-scale LP problem at each inner iteration, severely limiting the scalability and robustness of the framework. Such high magnitude of computations had prohibited it from many real-world applications until recently. To accelerate the computation of Wasserstein barycenter, and ultimately to improve D2-clustering, multiple numerical algorithmic efforts have been made in the recent few years (Cuturi and Doucet, 2014; Ye and Li, 2014; Benamou et al., 2015; Ye et al., 2017).

Although the effectiveness of Wasserstein distance has been well recognized in the computer vision and multimedia literature, the property of Wasserstein barycenter has not been well understood. To our knowledge, there still lacks systematic study of applying Wasserstein barycenter and D2-clustering in document analysis with word embeddings.

A closely related work to ours was proposed by authors of (Kusner et al., 2015) who recently connected the Wasserstein distance to the word embeddings for comparing documents. Our work differs from theirs in the methodology. We directly pursue a scalable clustering setting rather than construct a nearest neighbor graph based on calculated distances, because the calculation of the Wasserstein distances of all pairs is too expensive to be practical. The authors of (Kusner et al., 2015) used a lower bound that was cheaper to compute in order to prune unnecessary full distance calculation, but the scalability of this modified approach is still considered very limited, an issue to be discussed in Section 4.3. On the other hand, our approach adopts the framework similar to K-means which is of complexity $O(n)$ per iteration, and usually converges within tens of iterations. The computation of D2-clustering, though in its original form was magnitudes heavier than other document clustering methods, can now be done with efficient parallelization and proper implementations (Ye et al., 2017).

## 3 Our Approach

This section introduces the distance, the D2-clustering technique, the fast computation frame-

work, and how they are used in the proposed document clustering method.

## 3.1 Wasserstein Distance

Suppose we represent each document $d_k$ consisting $m_k$ unique words by a discrete measure or a discrete distribution, where $k = 1, \ldots, N$ with $N$ being the sample size:

$$d_k = \sum\nolimits_{i=1}^{m_k} w_i^{(k)} \delta_{x_i^{(k)}} \ . \tag{1}$$

Here $\delta_x$ denotes the Dirac measure with support $x$, and $w_i^{(k)} \geq 0$ is the "importance weight" for the $i$-th word in the $k$-th document, with $\sum_{i=1}^{m_k} w_i^{(k)} = 1$. And $x_i^{(k)} \in \mathbb{R}^d$, called a support point, is the semantic embedding vector of the $i$-th word. The 2nd-order Wasserstein distance between two documents $d_1$ and $d_2$ (and likewise for any document pairs) is defined by the following LP problem: $W^2(d_1, d_2) :=$

$$\begin{aligned} \min_{\Pi} \quad & \sum_{i,j} \pi_{i,j} \| x_i^{(1)} - x_j^{(2)} \|_2^2 \\ \text{s.t.} \quad & \sum_{j=1}^{m_2} \pi_{i,j} = w_i, \forall i, \quad \sum_{i=1}^{m_1} \pi_{i,j} = w_j, \forall j \\ & \pi_{i,j} \geq 0, \forall i,j \ , \end{aligned} \tag{2}$$

where $\Pi = \{\pi_{i,j}\}$ is a $m_1 \times m_2$ coupling matrix, and let $\{C_{i,j} := \| x_i^{(1)} - x_j^{(2)} \|_2^2\}$ be transportation costs between words. Wasserstein distance is a true metric (Villani, 2003) for measures, and its best exact algorithm has a complexity of $O(m^3 \log m)$ (Pele and Werman, 2009; Cuturi, 2013), if $m_1 = m_2 = m$.

## 3.2 Discrete Distribution (D2-) Clustering

D2-clustering (Li and Wang, 2008) iterates between the assignment step and centroids updating step in a similar way as the Lloyd's K-means. Suppose we are to find $K$ clusters. The assignment step finds each member distribution its nearest mean from $K$ candidates. The mean of each cluster is again a discrete distribution with $m$ support points, denoted by $c_i$, $i = 1, \ldots, K$. Each mean is iteratively updated to minimize its total within cluster variation. We can write the D2-clustering problem as follows: given sample data $\{d_k\}_{k=1}^N$, support size of means $m$, and desired number of clusters $K$, D2-clustering solves

$$\min_{c_1, \ldots, c_K} \sum_{k=1}^N \min_{1 \leq i \leq K} W^2(d_k, c_i) \ , \tag{3}$$

where $c_1, \ldots, c_K$ are Wasserstein barycenters. At the core of solving the above formulation is an optimization method that searches the Wasserstein barycenters of varying partitions. Therefore, we concentrate on the following problem. For each cluster, we reorganize the index of member distributions from $1, \ldots, n$. The Wasserstein barycenter (Agueh and Carlier, 2011; Cuturi and Doucet, 2014) is by definition the solution of

$$\min_c \sum_{k=1}^n W^2(d_k, c) \ , \tag{4}$$

where $c = \sum_{i=1}^m w_i \delta_{x_i}$. The above Wasserstein barycenter formulation involves two levels of optimization: the outer level finding the minimizer of total variations, and the inner level solving Wasserstein distances. We remark that in D2-clustering, we need to solve multiple Wasserstein barycenters rather than a single one. This constitutes the third level of optimization.

## 3.3 Modified Bregman ADMM for Computing Wasserstein Barycenter

The recent modified Bregman alternating direction method of multiplier (B-ADMM) algorithm (Ye et al., 2017), motivated by (Wang and Banerjee, 2014), is a practical choice for computing Wasserstein barycenters. We briefly sketch their algorithmic procedure of this optimization method here for the sake of completeness. To solve for Wasserstein barycenter defined in Eq. (4), the key procedure of the modified Bregman ADMM involves iterative updates of four block of primal variables: the support points of $c$ — $\{x_i\}_{i=1}^m$ (with transportation costs $\{C_{i,j}\}^{(k)}$ for $k = 1, \ldots, n$), the importance weights of $c$ — $\{w_i\}_{i=1}^m$, and two sets of split matching variables — $\{\pi_{i,j}^{(k,1)}\}$ and $\{\pi_{i,j}^{(k,2)}\}$, for $k = 1, \ldots, n$, as well as Lagrangian variables $\{\lambda_{i,j}^{(k)}\}$ for $k = 1, \ldots, n$. In the end, both $\{\pi_{i,j}^{(k,1)}\}$ and $\{\pi_{i,j}^{(k,2)}\}$ converge to the matching weight in Eq. (2) with respect to $d(c, d_k)$. The iterative algorithm proceeds as follows until $c$ converges or a maximum number of iterations are reached: given constant $\tau \geq 10$, $\rho \propto \frac{\sum_{i,j,k} C_{i,j}^{(k)}}{\sum_{k=1}^n m_k m}$ and round-off tolerance $\epsilon = 10^{-10}$, those variables are updated in the following order.

**Update** $\{x_i\}_{i=1}^m$ **and** $\{C_{i,j}^{(k)}\}$ **in every** $\tau$ **iterations:**

$$x_i := \frac{1}{n w_i} \sum_{k=1}^n \sum_{j=1}^{m_k} \pi_{i,j}^{(k,1)} x_j^{(k)}, \forall i, \tag{5}$$

$$C_{i,j}^{(k)} := \| x_i - x_j^{(k)} \|_2^2, \forall i, j \text{ and } k, \tag{6}$$

**Update** $\{\pi_{i,j}^{(k,1)}\}$ **and** $\{\pi_{i,j}^{(k,2)}\}$. For each $i, j$ and $k$,

$$\pi_{i,j}^{(k,2)} := \pi_{i,j}^{(k,2)} \exp\left(\frac{-C_{i,j}^{(k)} - \lambda_{i,j}^{(k)}}{\rho}\right) + \epsilon \quad (7)$$

$$\pi_{i,j}^{(k,1)} := w_j^{(k)} \pi_{i,j}^{(k,2)} \Big/ \left(\sum_{l=1}^{m} \pi_{l,j}^{(k,2)}\right) \quad (8)$$

$$\pi_{i,j}^{(k,1)} := \pi_{i,j}^{(k,1)} \exp\left(\lambda_{i,j}^{(k)}/\rho\right) + \epsilon \quad (9)$$

**Update** $\{w_i\}_{i=1}^{m}$. For $i = 1, \ldots, m$ ,

$$w_i := \sum_{k=1}^{n} \frac{\sum_{j=1}^{m_k} \pi_{i,j}^{(k,1)}}{\sum_{i,j} \pi_{i,j}^{(k,1)}} \quad (10)$$

$$w_i := w_i \Big/ \left(\sum_{i=1}^{m} w_i\right) \quad (11)$$

**Update** $\{\pi_{i,j}^{(k,2)}\}$ **and** $\{\lambda_{i,j}^{(k)}\}$. For each $i, j$ and $k$,

$$\pi_{i,j}^{(k,2)} := w_i \pi_{i,j}^{(k,1)} \Big/ \left(\sum_{l=1}^{m_k} \pi_{i,l}^{(k,1)}\right) \quad (12)$$

$$\lambda_{i,j}^{(k)} := \lambda_{i,j}^{(k)} + \rho\left(\pi_{i,l}^{(k,1)} - \pi_{i,l}^{(k,2)}\right) . \quad (13)$$

Eq. (5)-(13) can all be vectorized as very efficient numerical routines. In a data parallel implementation, only Eq. (5) and Eq. (10) (involving $\sum_{k=1}^{n}$) needs to be synchronized. The software package detailed in (Ye et al., 2017) was used to generate relevant experiments. We make available our codes and pre-processed datasets for reproducing all experiments of our approach.

# 4 Experiments

## 4.1 Datasets and Evaluation Metrics

We prepare six datasets to conduct a set of experiments. Two short-text datasets are created as follows. (1) BBCNews abstract: we concatenate the title and the first sentence of news posts from BBCNews dataset[1] to create an abstract version. (2) Wiki events: each cluster/class contains a set of news abstracts on the same story such as "2014 Crimean Crisis" crawled from Wikipedia current events following (Wu et al., 2015); this dataset offers more challenges because it has more fine-grained classes and fewer documents (with shorter length) per class than the others have. It also shows more realistic nature of real-world applications such as news event clustering.

We also experiment with two long-text datasets and two domain-specific text datasets.

---

[1]BBCNews and BBCSport are downloaded from http://mlg.ucd.ie/datasets/bbc.html

(3) Reuters-21578: we obtain the original Reuters-21578 text dataset and process as follows: remove documents with multiple categories, remove documents with empty body, remove duplicates, and select documents from the largest 10 categories. Reuters dataset is a highly unbalanced dataset (the top category has more than 3000 documents while the 10-th category has fewer than 100), this imbalance induces some extra randomness in comparing the results. (4) 20Newsgroups "bydate" version: we obtain the raw "bydate" versionand process them as follows: remove headers and footers, remove URLs and Email addresses, delete documents with less than 10 words. 20Newsgroups have roughly comparable sizes of categories. (5) BBCSports. (6) Ohsumed and Ohsumed-full: documents are medical abstracts from the MeSH categories of the year 1991. Specifically, there are 23 cardiovascular diseases categories.

Evaluating clustering results is known to be nontrivial. We use the following three sets of quantitative metrics to assess the quality of clusters by knowing the groundtruth categorical labels of documents: (1) Homogeneity, Completeness, and V-measure (Rosenberg and Hirschberg, 2007); (2) Adjusted Mutual Information (AMI) (Vinh et al., 2010); (3) Adjusted Rand Index (ARI) (Rand, 1971). For sensitivity analysis, we use the homogeneity score (Rosenberg and Hirschberg, 2007) as a projection dimension of other metrics, creating a 2D plot to visualize the metrics of a method along different homogeneity levels. Generally speaking, more clusters leads to higher homogeneity by chance.

## 4.2 Methods in Comparison

We examine four categories of methods that place a vector-space model over documents, and compare them to our D2-clustering framework. When needed, we use K-means++ to obtain clusters from dimension reduced vectors. To diminish the randomness brought by K-mean initialization, we ensemble the clustering results of 50 repeated runs (Strehl and Ghosh, 2003), and report the metrics for the ensembled one. The largest possible vocabulary used, excluding word embedding based approaches, is composed of words appearing in at least two documents. On each dataset, we select the same set of $K$s, the number of clusters, for all methods. Typically, $K$s are chosen around

the number of groundtruth categories in logarithmic scale.

We prepare two versions of the TF-IDF vectors as the unigram model. The ensembled K-means methods are used to obtain clusters. (1) *TF-IDF* vector (Sparck Jones, 1972). (2) *TF-IDF-N* vector is found by choosing the most frequent $N$ words in a corpus, where $N \in \{500, 1000, 1500, 2000\}$. The difference between the two methods highlights the sensitivity issue brought by the size of chosen vocabulary.

We also compare our approach with the following seven additional baselines. They are (3) *Spectral Clustering (Laplacian)* (4) *Latent Semantic Indexing (LSI)* (Deerwester et al., 1990). (5) *Locality Preserving Projection (LPP)* (He and Niyogi, 2004; Cai et al., 2005). (6) *Nonnegative Matrix Factorization (NMF)* (Lee and Seung, 1999; Xu et al., 2003). (7) *Latent Dirichlet Allocation (LDA)* (Blei et al., 2003; Hoffman et al., 2010; Pritchard et al., 2000). (8) *Average of word vectors (AvgDoc)* (9) *Paragraph Vectors (PV)* (Le and Mikolov, 2014). Details on their experimental setups and hyper-parameter search strategies can be found in the Appendix.

### 4.3 Runtime

We report the runtime for our approach upon two largest datasets. The experiments regarding other smaller datasets all finish within minutes in a single machine, which we omit due to page constraints. Like K-means, the running time spent by our approach depends on the number of actual iterations before a termination criterion is met. In the Newsgroups dataset, with $m = 100$ and $K = 45$, the time per iteration is 121 seconds on 48 processors. In Reuters dataset, with $m = 100$ and $K = 20$, the time per iteration is 190 seconds on 24 processors. Each run finishes in around tens of iterations typically, upon which the percentage of label changes is less than 0.1%.

Our approach adopts the Elkan's algorithm pruning unnecessary computations of Wasserstein distance in assignment steps of K-means (Elkan, 2003). For the Newsgroups data (with $m = 100$ and $K = 45$), our approach terminates in 36 iterations, and totally computes $12,162,717$ ($\approx 3.5\% \times 18612^2$) distance pairs in assignment steps, saving 60% ($\approx 1 - \frac{12,162,717}{36 \times 45 \times 18612}$) distance pairs to calculate in the standard D2-clustering. In comparison, the clustering approaches based on

| Method | | EMD counts |
|---|---|---|
| Our approach | $d = 400, K = 10$ | 2.0 |
| Our approach | $d = 400, K = 40$ | 3.5 |
| KNN | $d = 400, K = 1$ | 73.9 |
| KNN | $d = 100, K = 1$ | 53.0 |
| KNN | $d = 50, K = 1$ | 23.4 |

Table 1: Percentage of total $18612^2$ Wasserstein distance pairs needed to compute on the full Newsgroup dataset. The KNN graph based on 1st order Wasserstein distance is computed from the prefetch-and-prune approach according to (Kusner et al., 2015).

K nearest neighbor graph with the prefetch-and-prune method of (Kusner et al., 2015) needs substantially more pairs to compute Wasserstein distance, meanwhile the speed-ups also suffer from the curse of dimensionality. Their detailed statistics are reported in Table 1. Based on the results, our approach is much more practical as a basic document clustering tool.

### 4.4 Results

We summarize our numerical results in this section.

| Dataset | size | class | length | est. #voc. |
|---|---|---|---|---|
| BBCNews abstr. | 2,225 | 5 | 26 | 7,452 |
| Wiki events | 1,983 | 54 | 22 | 5,313 |
| Reuters | 7,316 | 10 | 141 | 27,792 |
| Newgroups | 18,612 | 20 | 245 | 55,970 |
| BBCSports | 737 | 5 | 345 | 13,105 |
| Ohsumed | 4,340 | 23 | - | - |
| Ohsumed-full* | 34,386 | 23 | 184 | 43,895 |

Table 2: Description of corpus data that have been used in our paper. *Ohsumed-full dataset is used for pre-training word embeddings only. Ohsumed is a downsampled evaluation set resulting from removing posts from Ohsumed-full that belong to multiple categories.

**Regular text datasets**. The first four datasets in Table 2 cover quite general and broad topics. We consider them to be regular and representative datasets encountered more frequently in applications. We report the clustering performances of the ten methods in Fig. 1, where three different metrics are plotted against the clustering homogeneity. The higher result at the same level of homogeneity is better, and the ability to achieve higher homogeneity is also welcomed. Clearly, D2-clustering is the only method that shows ro-

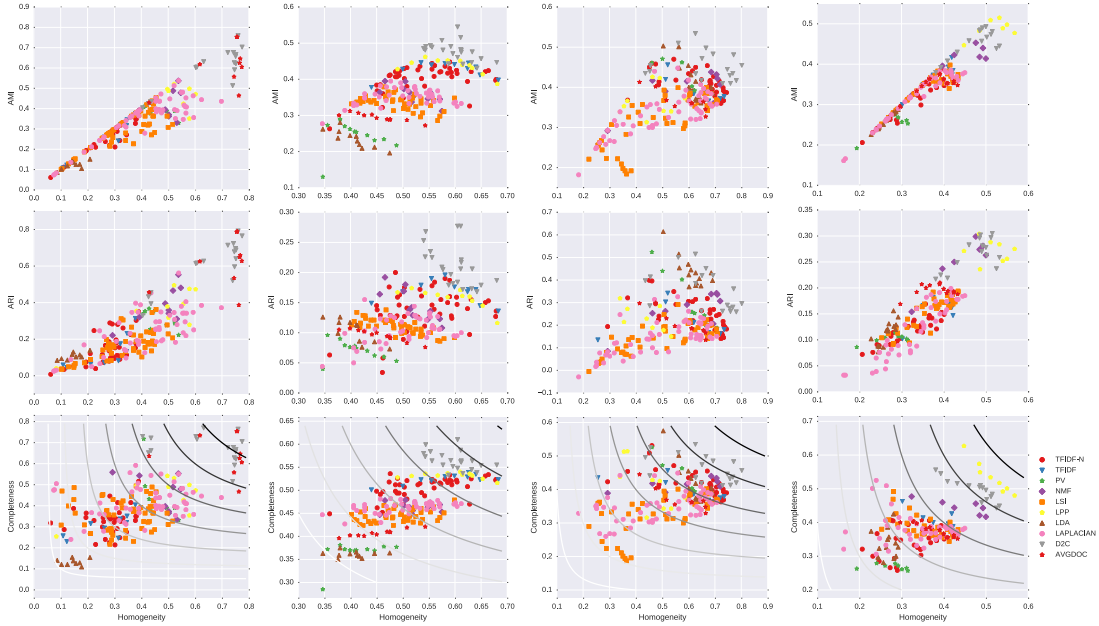

Figure 1: The quantitative cluster metrics used for performance evaluation of "BBC title and abstract","Wiki events", "Reuters", and "Newsgroups" (column-wise, from left to right): y-axis corresponds to AMI, ARI, and Completeness respective (row-wise, from top to down), and x-axis corresponds to Homogeneity for sensitivity analysis.

bustly superior performances among all ten methods. Specifically, it ranks first in three datasets, and second in the other one. In comparison, LDA performs competitively on the "Reuters" dataset, but is substantially unsuccessful on others. Meanwhile, LPP performs competitively on the "Wiki events" and "Newsgroups" datasets, but it underperforms on other two. Laplacian, LSI, and Tfidf-N can achieve comparably performance if their reduced dimensions are fine tuned, which unfortunately is not realistic in practice. NMF is a simple and effective method which always gives stable, though subpar, performance.

**Short texts vs. long texts**. D2-clustering performs much more impressive on short texts ("BBC abstract" and "Wiki events") than it does on long texts ("Reuters" and "Newsgroups"). This outcome is somewhat expected, because the bag-of-words method suffers from high sparsity for short texts, and word-embedding based methods in theory should have an edge here. As shown in Fig. 1, D2-clustering has indeed outperformed other non-embedding approaches by a large margin on short texts (improved by about 40% and 20% respectively). Nevertheless, we find lifting from word embedding to document clustering is not a free

lunch. Neither AvgDoc nor PV can perform as competitively as D2-clustering performs on both.

**Domain-specific text datasets**. We are also interested in how word embedding can help group domain-specific texts into clusters. In particular, does the semantic knowledge "embedded" in words provides enough clues to discriminate fine-grained concepts? We report the best AMI achieved by each method in Table 3. Our preliminary result indicates state-of-the-art word embeddings do not provide enough gain here to exceed the performance of existing methodologies. On the easy one, aka "BBCSport" dataset, basic bag-of-words approach (Tfidf and Tfidf-N) already suffices to discriminate different sport categories; and on the hard one, aka "Ohsumed" dataset, D2-clustering only slightly improves over Tf-idf and others, ranking behind LPP. Meanwhile, we feel the overall quality of clustering "Ohsumed" texts is quite far from useful in practice, no matter which method to use. (See next section for more discussions.)

## 4.5 Sensitivity to Word Embeddings.

We validate the robustness of D2 clustering with different word embedding models, and we also

|  | regular dataset | | | | domain-specific dataset | | |
|---|---|---|---|---|---|---|---|
|  | BBCNews abstract | Wik events | Reuters | Newsgroups | BBCSport | Ohsumed | Avg. |
| Tfidf-N | 0.389 | 0.448 | 0.470 | 0.388 | **0.883** | 0.210 | 0.465 |
| Tfidf | 0.376 | 0.446 | 0.456 | 0.417 | 0.799 | 0.235 | 0.455 |
| Laplacian | 0.538 | 0.395 | 0.448 | 0.385 | 0.855 | 0.223 | 0.474 |
| LSI | 0.454 | 0.379 | 0.400 | 0.398 | 0.840 | 0.222 | 0.448 |
| LPP | 0.521 | 0.462 | 0.426 | **0.515** | 0.859 | **0.284** | 0.511 |
| NMF | 0.537 | 0.395 | 0.438 | 0.453 | 0.809 | 0.226 | 0.476 |
| LDA | 0.151 | 0.280 | 0.503 | 0.288 | 0.616 | 0.132 | 0.328 |
| AvgDoc | **0.753** | 0.312 | 0.413 | 0.376 | 0.504 | 0.172 | 0.422 |
| PV | 0.428 | 0.289 | 0.471 | 0.275 | 0.553 | 0.233 | 0.375 |
| D2C (Our approach) | **0.759** | **0.545** | **0.534** | 0.493 | 0.812 | 0.260 | **0.567** |

Table 3: Best AMIs (Vinh et al., 2010) of compared methods on different datasets and their averaging. The best results are marked as bold font for each dataset, the 2nd and the 3rd are marked by blue and magenta colors respectively.

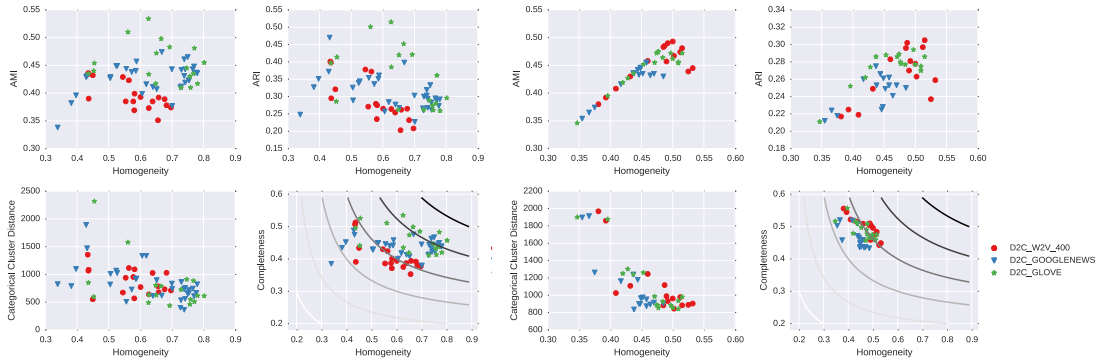

Figure 2: Sensitivity analysis: the clustering performances of D2C under different word embeddings. Left: Reuters, Right: Newsgroups. An extra evaluation index — categorical cluster distance (Zhou et al., 2005) is also used.

show all their results in Fig. 2. As we mentioned, the effectiveness of Wasserstein document clustering depends on how relevant the utilized word embeddings are with the tasks. In those general document clustering tasks, however, word embedding models trained on general corpus perform robustly well with acceptably small variations. This outcome reveals our framework as generally effective and not dependent on a specific word embedding model. In addition, we also conduct experiments with word embeddings with a smaller dimension, aka 50 and 100. Their results are not as good as those we reported (therefore detailed numbers are not included due to space limit).

**Inadequate embeddings may not be disastrous**. In addition to our standard running set, we also try D2-clustering with purely random word embeddings, meaning each word vector is independently sampled from spherical Gaussian at 300 dimen-

|  | ARI | AMI | V-measure |
|---|---|---|---|
| BBCNews abstract | .146 | .187 | .190 |
|  | $.792_{+442\%}$ | $.759_{+306\%}$ | $.762_{+301\%}$ |
| Wiki events | .194 | .369 | .463 |
|  | $.277_{+43\%}$ | $.545_{+48\%}$ | $.611_{+32\%}$ |
| Reuters | .498 | .524 | .588 |
|  | $.515_{+3\%}$ | $.534_{+2\%}$ | $.594_{+1\%}$ |
| Newsgroups | .194 | .358 | .390 |
|  | $.305_{+57\%}$ | $.493_{+38\%}$ | $.499_{+28\%}$ |
| BBCSport | .755 | .740 | .760 |
|  | $.801_{+6\%}$ | $.812_{+10\%}$ | $.817_{+8\%}$ |
| Ohsumed | .080 | .204 | .292 |
|  | $.116_{+45\%}$ | $.260_{+27\%}$ | $.349_{+20\%}$ |

Table 4: Comparison between *random* word embeddings (upper row) and meaningful *pre-trained* word embeddings (lower row), based on their best ARI, AMI, and V-measures. The improvements by percentiles are also shown in the subscripts.

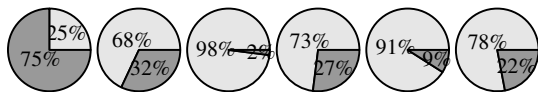

Figure 3: Pie charts of clustering gains in AMI calculated from our framework. Light region is by bag-of-words, and dark region is by pre-trained word embeddings. Six datasets (from left to right): BBCNews abstract, Wiki events, Reuters, Newsgroups, BBCSport, and Ohsumed.

sion, to see how deficient it can be. From our experimental results, random word embeddings degrade the performance of D2-clustering, but it still performs much better than purely random clustering does, and is even consistently better than LDA. Its performances across different datasets is highly correlated with the bag-of-words (Tfidf and Tfidf-N). By comparing a pre-trained word embedding model to a randomly generated one, we find that the extra gain is significant ($> 10\%$) in clustering four of the six datasets. Their detailed statistics are in Table 4 and Fig. 3.

## 5 Discussions

**Performance advantage**. There has been one immediate observation from these studies, D2-clustering always outperforms two of its degenerated cases, namely Tf-idf and AvgDoc, and three other popular methods: LDA, NMF, and PV, on all tasks. Therefore, for document clustering, users can expect to gain performance improvements by using our approach.

**Clustering sensitivity**. From the four 2D plots in Fig. 1, we notice that the results of Laplacian, LSI and Tfidf-N are rather sensitive to their extra hyper-parameters. Once the vocabulary set, weight scheme and embeddings of words are fixed, our framework involves only two additional hyper-parameters: the number of intended clusters, $K$, and the selected support size of centroid distributions, $m$. We have chosen more than one $m$ in all related experiments ($m = \{64, 100\}$ for long documents, and $m = 10, 20$ for short documents). Our empirical experiments show that the effect of $m$ on different metrics is less sensitive than the change of $K$. As shown in Fig. 1, results at different $K$ are plotted for each method. The gray dots denote results of multiple runs of D2-clustering. They are always contracted around the top-right region of the whole population, revealing

the predictive and robustly supreme performance of our approach.

**When bag-of-words suffices**. Among the results of "BBCSport" dataset, Tfidf-N shows that by restricting the vocabulary set into a smaller one (which may be more relevant to the interest of tasks), it already can achieve highest clustering AMI without any other techniques. Other unsupervised regularization over data is likely unnecessary, or even degrades the performance slightly.

**Toward better word embeddings**. Our experiments on the Ohsumed dataset have been limited. The result shows that it could be highly desirable to incorporate certain domain knowledge to derive more effective vector embeddings of words and phrases to encode their domain-specific knowledge, such as jargons that have knowledge dependencies and hierarchies in educational data mining, and signal words that capture multi-dimensional aspects of emotions in sentiment analysis.

Finally, we report the best AMIs of all methods on all datasets in Table 3. By looking at each method and the average of best AMIs over six datasets, we find our proposed clustering framework often performs competitively and robustly, which is the only method reaching more than 90% of the best AMI on each dataset. Furthermore, this observation holds for varying lengths of documents and varying difficulty levels of clustering tasks. Our nonparametric framework benefits from both bag-of-words and word embeddings.

## 6 Conclusions

This paper introduces a nonparametric clustering framework for document analysis. Its computational tractability, robustness and supreme performance, as a fundamental tool, are empirically validated. Its ease of use enables data scientists to use it for the pre-screening purpose of examining word embeddings in a specific task. Finally, the gains acquired from word embeddings are quantitatively measured from a nonparametric unsupervised perspective.

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
