# Peer review of "Determining Gains Acquired from Word Embedding Quantitatively Using Discrete Distribution Clustering"

_ACL 2017 — decision unknown_

[Official Review · Reviewer 1 · rating 4 · confidence 4]
soundness 4 · originality 3 · clarity 3 · impact 3 · substance 2 · appropriateness 4 · meaningful comparison 3 · presentation format Poster

- Strengths:

- Weaknesses:
Many grammar errors, such as the abstract

- General Discussion:

[Official Review · Reviewer 2 · rating 4 · confidence 4]
soundness 4 · originality 3 · clarity 4 · impact 3 · substance 4 · appropriateness 5 · meaningful comparison 3 · presentation format Oral Presentation

- Strengths: Introduces  a new document clustering approach and compares it to
several established methods, showing that it improves results in most cases.
The analysis is very detailed and thorough--quite dense in many places and
requires careful reading.

The presentation is organized and clear, and I am impressed by the range of
comparisons and influential factors that were considered. Argument is
convincing and the work should influence future approaches.

- Weaknesses:

 The paper does not provide any information on the availability of the software
described.

- General Discussion:

Needs some (minor) editing for English and typos--here are just a few:

Line 124: regardless the size > regardless of the size
Line 126: resources. Because > resources, because
Line 205: consist- ing mk > consisting of mk
Line 360: versionand > version and